# Aberrant B Cell Signaling in Autoimmune Diseases

**DOI:** 10.3390/cells11213391

**Published:** 2022-10-27

**Authors:** Odilia B. J. Corneth, Stefan F. H. Neys, Rudi W. Hendriks

**Affiliations:** Department of Pulmonary Medicine, Erasmus MC, University Medical Center, 3015 GD Rotterdam, The Netherlands

**Keywords:** autoimmunity, B cell, B cell receptor, kinase, receptor, Toll-like receptor, signal transduction, tolerance

## Abstract

Aberrant B cell signaling plays a critical in role in various systemic and organ-specific autoimmune diseases. This is supported by genetic evidence by many functional studies in B cells from patients or specific animal models and by the observed efficacy of small-molecule inhibitors. In this review, we first discuss key signal transduction pathways downstream of the B cell receptor (BCR) that ensure that autoreactive B cells are removed from the repertoire or functionally silenced. We provide an overview of aberrant BCR signaling that is associated with inappropriate B cell repertoire selection and activation or survival of peripheral B cell populations and plasma cells, finally leading to autoantibody formation. Next to BCR signaling, abnormalities in other signal transduction pathways have been implicated in autoimmune disease. These include reduced activity of several phosphates that are downstream of co-inhibitory receptors on B cells and increased levels of BAFF and APRIL, which support survival of B cells and plasma cells. Importantly, pathogenic synergy of the BCR and Toll-like receptors (TLR), which can be activated by endogenous ligands, such as self-nucleic acids, has been shown to enhance autoimmunity. Finally, we will briefly discuss therapeutic strategies for autoimmune disease based on interfering with signal transduction in B cells.

## 1. Introduction

B lymphocytes have the unique capacity to recognize pathogen-derived antigens through expression of the B cell receptor (BCR) on their cell surface. However, the random nature of the VDJ-recombination process that generates antibody diversity poses the potential danger of producing self-reactive B cells that ultimately contribute to an autoimmune response. Autoreactive B cells play a crucial role in the pathogenesis of various common systemic and organ-specific autoimmune diseases, including systemic lupus erythematosus (SLE), rheumatoid arthritis (RA), Sjögren’s syndrome (SjS), type 1 diabetes (T1D), cutaneous autoimmune diseases (CAD), and multiple sclerosis (MS). The autoantibodies produced are typically involved in immune complex formation and deposit in target organs. Autoantibodies often appear in the serum many years before clinical disease onset, suggesting that an early breach of B cell tolerance contributes to autoimmune pathogenesis [1,2]. However, the stages of B cell differentiation that account for the breach of self-tolerance or the underlying mechanisms remain largely unknown, as do the numerous genetic and environmental factors at play.

Genome-wide association studies (GWAS) have uncovered many polymorphisms that are associated with autoimmune disorders, although it has been quite challenging to obtain mechanistic insight from these genetic studies [3,4,5]. For example, more than 30 loci have been identified that show robust association with SLE [6]. These SLE susceptibility genes tend to cluster in Toll-like receptor (TLR), BCR, or Fc-receptor signaling pathways, immune-complex processing, or antigen presentation. Interestingly, SLE susceptibility loci, such as *BLK*, *STAT4*, *TNFAIP3*, *BANK1*, [7], and *PTPN22* [8], have also been implicated in other (systemic) autoimmune diseases.

A critical role for aberrant B cell signaling pathways in autoimmunity is not only supported by GWAS, but also by numerous functional studies using B cells from patients, animal models, or by studies using small-molecule inhibitors. In this review, we evaluate how BCR signaling ensures that autoreactive B cells are removed or silenced during B cell development. Next, we provide an overview of inadvertent B cell activation and the aberrant signaling downstream of the BCR and various other receptors observed in autoimmune disease. Therapeutic strategies for autoimmune disease based on interfering with signal transduction in B cells have been the topic of many recent reviews [9,10,11,12] and will be briefly discussed.

## 2. Shaping of the Naïve B Cell Repertoire during B Cell Development by B Cell Receptor Signals

A diverse antigen receptor repertoire is generated by the unique stochastic process of DNA recombination, mediated by the recombinase-activating gene (RAG) proteins. Hereby, gene segments encoding variable (V), diversity (D), and joining (J) regions of the immunoglobulin (Ig) heavy (H) and light (L) chain loci are assembled in a stepwise fashion [13,14]. Upon expression of a functionally rearranged IgH chain in early B cell precursors, a complex interplay between pre-BCR, interleukin (IL)-7 receptor (IL-7R) and chemokine receptor CXCR4 signaling induces pre-B cell proliferation and subsequent IgL chain recombination (Figure 1A, 1) [15,16]. Hereby, phosphoinositide 3-kinase (PI3K) signaling is required for pre-B cell survival [17], but negative selection of pre-B cells expressing a strongly autoreactive pre-BCR may be facilitated through hyperactivation of the PI3K-AKT pathway, as this leads to metabolic stress and AKT-dependent cell death [18,19]. Following successful IgL chain recombination in small pre-B cells, the high levels of CXCR4 surface expression are sharply reduced, facilitating the export of surface IgM-expressing immature B cells into bone marrow (BM) sinusoids and egress into the circulation [20].

It has been estimated that up to ~75% of B cells that develop in the BM display some degree of autoreactivity and must, therefore, be removed from the repertoire [21]. This frequency was assessed by determining the reactivity of cloned and in vitro amplified recombinant antibodies derived from human single B cells from BM. The regulatory mechanism that reduces the frequency of self-reactivity at the immature B cell stage in the BM is referred to as central tolerance and is driven by BCR signals. BCR engagement by self-antigens prevents CXCR4 downregulation, increases the motility and retention of immature B cells within the BM parenchyma, and blocks their egress [20]. Recognition of self-antigens will induce secondary gene rearrangement at the IgL chain loci. This process is termed receptor editing and modifies the specificity of potentially harmful BCRs (Figure 1A, 2). Whereas receptor editing is particularly frequent in IgM^lo^ early immature B cells, the subsequent stage of IgM^hi^ immature B cells is highly sensitive to antigen-induced apoptosis [22]. Infection or immunization may suppress lymphopoiesis in the BM and can subsequently result in antigen-independent accumulation of RAG-expressing immature B cells in the spleen [23]. Such transient alterations in lymphopoiesis are thought to protect against tolerance and to indirectly enhance B cell memory. Moreover, newly formed B cells that emerge from the BM will have a different repertoire and may, therefore, respond to pathogens in a distinct manner [24]. Interestingly, B cells that develop in the intestinal lamina propria are also subject to receptor editing, resulting in a BCR repertoire that is shaped by extracellular signals from commensal microbes [25].

A substantial proportion of self-reactive B cells (~40%) leaves the BM and further mechanisms ensure that these B cells are kept in check. In contrast to the BM that provides a protective micro-environment for immature B cells that allows for receptor editing, recent new immigrant B cells, called transitional B cells, in the spleen are largely eliminated when recognizing self-antigen (Figure 1B, 3) [26]. Nevertheless, using transgenic mice harboring a green fluorescent reporter, it was shown that a small fraction of transitional B cells in the spleen had an IgM^lo^ phenotype and did not terminate RAG expression [27]. Although these cells express substantially lower levels of RAG than immature B cells in the BM [27,28], they had detectable levels of DNA double-strand breaks. Therefore, receptor editing events in recent BM emigrants may continue to some extent [27].

The high sensitivity of transitional B cells to BCR signaling-induced apoptosis reduces self-reactivity of B cells to ~20% [21,29]. Thus, a significant change in BCR repertoire occurs as transitional B cells enter the stage of long-lived mature B cells [30]. Alternatively, self-reactive B cells may persist in the periphery but are functionally silenced by anergy, which is induced by chronic BCR signaling and the ensuing feedback loops (Figure 1B, 4) [31,32]. B cells recognizing autoantigens are poorly competitive with non-autoreactive naïve B cells to capture B-cell-activating factor (BAFF) [33,34], a tumor necrosis factor (TNF) family member that critically enhances B cell survival [35]. Accordingly, pharmacological inhibition of BAFF by belimumab, the first biological approved for SLE, was shown to be clinically effective in patients [36]. Even if B cells have been selected into the long-lived pool of peripheral B cells, expression of an autoreactive BCR leads to their rapid elimination. This was shown in elegant mouse experiments using inducible Cre-loxP-mediated gene inversion that changed BCR specificity [37].

## 3. Activation of Self-Reactive B Cells under Physiological Conditions

Antigen-activated B cells may undergo clonal expansion, IgH chain class switch recombination (CSR), affinity maturation by somatic hypermutation (SHM), and final differentiation into either memory B cells or antibody-secreting cells. Although it was generally assumed that CSR mainly takes place in germinal centers (GCs), isotype switching has been detected early after B cell activation and in extrafollicular responses [38]. Indeed, convincing evidence was provided that CSR induction precedes GC B cell differentiation and that the majority of CSR events occur outside the GCs prior to the onset of SHM [39]. SHM can increase BCR affinity towards antigens but, at the same time, poses the risk for de novo generation of autoreactive B cells. Only B cells with high antigen affinity that interact with T-helper cells specific for the same antigen will be selected during the GC reaction. GC B cells with affinity for a self-antigen lack proper T cell help and are subject to negative selection (Figure 1B, 5). This is based on the upregulation of the death receptor FAS (CD95; TNF family receptor 6) when B cells are activated by BCR engagement and CD40–CD40L interactions [40]. FAS signaling induces programed cell death upon interaction with its ligand on activated T cells. The main T cell subset that controls GC B cell selection is the follicular T helper (Tfh) cells, which are characterized by surface expression of chemokine receptor CXCR5 and programed cell death-1 (PD-1), and by production of IL-21 [41,42,43]. Recently, a CXCR5^+^PD-1^+^ subpopulation of CD8 T cells was identified, which also regulates the GC B cell response and B cell tolerance [44]. Repression of unwanted Tfh and GC B cell activity and promotion of stringent high-affinity B cell selection is further reinforced by FoxP3^+^ follicular regulatory T (Tfr) cells (Figure 1B) [45,46,47].

Apart from the critical role of the BCR signaling cascade in B cell activation following antigen encounter, sustained low-level BCR signaling, also referred to as tonic BCR signaling, is required for survival of both developing and mature B cells [17,48]. For specific B cells, it has been demonstrated that the presence of self-antigen provides a survival signal. In a transgenic mouse model in which B cells carry an autoreactive BCR recognizing the Thy-1 (CD90) glycoprotein, the presence of self-antigen promoted the accumulation of Thy-1-specific B-1 cells in the peritoneal cavity [49] and directed maturation of naïve immature B cells into the marginal zone B cell subset (Figure 1B) [50]. Given that B-1 cells can be selected and maintained on the basis of their autoreactivity, it is expected that (part of) the natural antibodies in serum will be the product of a self-antigen-driven process. Accordingly, in addition to the role of natural antibodies in early antimicrobial host defense, they facilitate nonimmunogenic clearance of apoptotic cells, removal of self-antigens, and inhibit responses induced by IgG autoantibodies [51,52,53]. In this context, it is of note that autoreactive IgM antibodies that recognize insulin were very recently shown to act as key regulators of blood glucose and metabolism [54]. These antibodies control the concentration of insulin in blood: whereas low-affinity anti-insulin IgM neutralizes insulin and leads to increased blood glucose, high-affinity anti-insulin IgM protects insulin neutralization by anti-insulin IgG. Consistent with these findings, antibody-deficient mice or immunodeficiency patients have sub-physiological blood glucose concentrations. The phenomenon that IgM autoantibodies have the capacity to prevent autoimmune pathology by competing with self-destructive autoantibodies was coined adaptive tolerance [55]. Intriguingly, these findings imply that self-reactive or poly-reactive B cells, present in the circulation of healthy individuals, have functional relevance beyond a role in host defense.

## 4. Defective Selection of Self-Reactive B Cells in Autoimmune Disease

Autoantibodies may appear in the circulation of patients many years before the onset of the clinical disease symptoms, suggesting that a break in B cell tolerance is an early event in the pathogenesis of autoimmune disease [1,56,57,58,59]. However, controversy exists regarding the stages of B cell development or activation that are defective in autoimmune disease. Patients may suffer from intrinsic B cell defects that hamper counterselection of autoreactive cells in early tolerance checkpoints or later checkpoints upon antigen-driven activation and differentiation in the GC response. Moreover, cell-intrinsic defects can affect specific B cell populations, such as anergic B cells or age-associated B cells (ABCs; described below). Finally, autoimmune pathology may result from dysregulated T cell help or regulatory T cells. It is thought that the controversy in the literature can be explained by major differences between individual autoimmune diseases and a large heterogeneity across patients with the same disease. This heterogeneity is also reflected by differences across the many spontaneous, induced, genetically modified, or humanized mouse models available for human autoimmune diseases, such as SLE, RA, and SjS [60,61,62]. In the coming section, we will provide a brief overview of the main defects reported in autoimmune patients and mouse models.

It is conceivable that defects in early tolerance checkpoints in the BM or spleen result in the accumulation of autoreactive naïve mature B cells in the circulation of patients with autoimmune disease [63]. These B cells may induce or promote autoimmunity because of their capacity to present self-antigens to T cells. Frequency analysis of antinuclear antibody (ANA)-expressing B cells in two classic lupus-prone mouse strains revealed heterogeneity regarding the B cell stage in which tolerance is first breached [64]. In the MRL/*lpr* mouse, one of the best characterized models for SLE, survival of autoreactive B and T cells is enhanced due to the recessive autosomal *lpr* mutation that results in defective FAS expression [60]. In BM and spleen of MRL/*lpr* mice, proportions of ANA^+^ B cells were similar across all B cell subsets and were in the range of nonautoimmune strains, indicating that the early tolerance checkpoints were intact. In contrast, NZB/W mice displayed an increase in ANA^+^ naïve mature B cells, suggesting a defect in early pre-immune tolerance [64]. Interestingly, both mouse strains did not display an increase in the proportions of ANA^+^ IgG-switched memory B cells and plasmablasts. Rather, a general expansion of switched memory cells resulted in increased numbers of ANA^+^ antigen-experienced cells.

Likewise, the frequencies of ANA^+^ cells within the circulating populations of IgG_1_-switched memory B cells or plasmablasts in SLE patients were reported to be similar to healthy controls [64]. Nevertheless, an increase in the total number of ANA^+^ IgG_1_^+^ plasma cells may be due to an overall expansion of the IgG_1_^+^ plasma cell compartment. These findings indicated that SLE does not reflect a defect in antigen-specific B cell tolerance but results from a generalized aberrant late B cell differentiation. They contradict earlier studies that described BCR repertoire abnormalities in SLE, including differences in IgV gene usage and IgH chain complementarity-determining region-3 characteristics [65]. Analysis of >200 cloned and in vitro expressed antibodies from single human B cells from three SLE patients revealed a ~twofold increase in autoreactivity in the mature naïve B cell population of SLE patients, compared with healthy controls [66]. No significant increase in autoreactive antibodies was found in new immigrant B cells. The level of poly-reactivity was increased in both new immigrant B cells and mature naïve B cells, but counterselection differed substantially between patients. SLE patients in clinical remission also showed elevated numbers of self-reactive or poly-reactive mature naïve B cells, indicating that early checkpoint abnormalities are an integral feature of SLE, regardless of disease status [67]. Taken together, these findings illustrate that, in different SLE patients, different checkpoints are affected.

Similar BCR repertoire analyses revealed defective central B cell tolerance in patients with RA, T1D, SjS, myasthenia gravis, and neuromyelitis optica spectrum disease [68]. Frequencies of self-reactive or poly-reactive transitional B cells in the circulation of these patient groups were increased, compared with healthy donors. These findings suggest that peripheral B cell tolerance checkpoints are disturbed [68]. In contrast, whereas five out of seven patients with MS displayed unaltered central tolerance, in all seven patients, analyzed peripheral tolerance was hampered. These findings point to a distinct B cell defect in MS.

Evidence for a major role of BCR signaling in shaping the repertoire of naïve B cells is provided by the finding of defective central tolerance in patients with mutations in essential signaling components, such as Bruton’s tyrosine kinase (*BTK*) and Wiskott–Aldrich syndrome protein (*WASP*) genes [69,70]. Moreover, TLR signaling critically contributes to central B cell tolerance, given the defective removal of developing autoreactive B cells in patients with mutations in genes encoding myeloid differentiation primary response 88 (MyD88), IL-1R-associated kinase-4 (IRAK-4), and the UNC-98 chaperone [69,70,71]. Nevertheless, these patients display an immunodeficiency rather than an autoimmune disease because their B cells are either almost completely lacking (in case of BTK deficiency) or are not activated due to the underlying mutation. Transitional and mature naïve B cells from patients or mice deficient for activation-induced cytidine deaminase (AID), which mediates CSR and SHM in B cells, also express an abnormal Ig repertoire that is associated with impaired B cell tolerance [72,73]. The mechanisms involved are not well defined but are thought to be B-cell-intrinsic and to depend on AID expression at the immature B cell stage in the BM [74]. In particular, it has been shown in mice that BCR and endosomal TLR signals synergize to induce high AID expression in immature B cells to levels that approach those of GC B cells. Such AID-expressing immature B cells lack antiapoptotic Mcl-1 and are normally deleted by apoptosis [75,76].

## 5. Defective Activation of Self-Reactive B Cells in Autoimmune Disease

### 5.1. Enhanced Sensitivity of Naïve B Cells for Activating Signals

Although autoimmunity is generally associated with BCR repertoire changes and enhanced persistence of autoreactive B cells in the circulation [77], it is becoming increasingly clear that, in autoimmune patients, naïve B cells probably need fewer strong signals to be activated. Analyses of transcriptomes, methylomes, and chromatin accessibility by Scharer et al. unraveled an SLE-specific epigenomic signature in resting naïve B cells [78,79]. This signature was characterized by increased enrichment of accessible chromatin at loci surrounding genes involved in B cell activation. Sites of open chromatin were enriched for motifs for AP-1 and EGR transcription factors, which have been linked to autoimmunity and are induced by BCR engagement [78,79]. In addition, naïve B cells displayed increased accessibility at the nuclear receptor subfamily 4 members *NRA4A1* (NUR77) and *NRA4A3*, which are known to be induced in response to BCR and TLR stimulation, respectively. It is thought that the epigenetic profile of naïve B cells is a reflection of both genetic and specific environmental factors. The latter may include unique signals provided by an autoimmune micro-environment, (proinflammatory) serum factors, or interactions with other cells of the immune system, for example, direct dendritic cell–B cell interplay [80]. However, the nature of these signals needs to be further explored.

### 5.2. Activation of Anergic B Cells

Anergy contributes to tolerance but, when anergic self-reactive B cells are reactivated, they may induce autoimmune pathology. Anergic B cells are impaired in their activation, proliferation, and differentiation into plasma cells and have a short life span, particularly when they are competing with non-anergic B cells. Their anergic state is (i) dependent on continuous (self)-antigen binding; (ii) associated with downregulation of surface IgM expression, but not IgD, and enhanced BCR endocytosis; and (iii) involves inhibitory co-receptors that recruit protein phosphatases [32]. In addition, chronic BCR stimulation induces changes in transcription factor expression and epigenetic modifications, which contribute to the anergic state. A defect in anergic B cells has been shown to be an important pathogenic mechanism that contributes to autoimmunity (as reviewed by Franks and Camier [81]). Though it remains largely unclear whether this originates from hampered anergy induction, inappropriate activation of anergic B cell, or both, studies indicate T-helper signals may be responsible for restoring BCR signaling in autoreactive anergic B cells [82,83].

Interestingly, anergic B cells should not be regarded as only potentially dangerous cells that need to be fully silenced. For example, they are advantageous, as their reactivation allows for the generation of broadly neutralizing antibodies (bnAbs) for HIV and influenza virus that have poly-reactive or autoreactive specificities [84,85]. Sequence analysis of isotype-switched memory B cells or somatically mutated bnAbs indicated that the naïve B cells expressing unmutated equivalents of these BCRs were likely anergic. It is assumed that their self-reactivity is removed or their affinity for foreign antigens is enhanced by SHM and selection in GCs in a process called clonal redemption (Figure 1B) [32,86]. Due to a ‘redeeming’ somatic mutation, the chronic anergy-inducing BCR stimulation can change into a tonic-like or antigen-stimulated BCR signal [87].

### 5.3. GC Selection Defects and Spontaneous GC Formation

Given that ANAs in SLE and anticitrullinated protein antibodies (ACPAs) in RA are class-switched and highly somatically hypermutated, it is generally thought that they are GC-derived. This is supported by several findings. It was reported that GC exclusion of autoreactive B cells is defective in human SLE [88]. Moreover, the chemokine receptor CXCR4, which is critical for segregating GC dark and light zone and B cell selection, was significantly upregulated in B cells from SLE patients and positively correlated with disease activity [89]. In systemic autoimmune disease, including SLE, SjS, and RA, an increase in circulating Tfh cells was identified, which correlated with disease activity or autoantibody titers [90,91,92], and normalized upon B cell depletion therapy [93]. Typically, ectopic GC formation is observed in inflamed tissues, including synovial tissue in RA, lacrimal and salivary glands in SjS, and the meninges of patients with progressive MS, where they act as local centers of the autoimmune response and autoantibody production [94]. Moreover, lymphatic aggregates in the skin, named skin-associated lymphatic tissue, may act as a niche for localized autoantibody production in CAD, such as pemphigus [11].

Activation of autoreactive B cells and the generation of class-switched pathogenic antibodies is thought to occur in spontaneous GCs, in the absence of immunization or any detectable infection, and essentially independent of commensal microbiota [95]. However, it cannot be excluded that endogenous viruses or retroviral elements play a role (reviewed by Domeier er al. [96]). Spontaneous GCs are observed in a wide range of autoimmune mouse strains, particularly in mice with altered signaling or survival of B cells, for example, due to BAFF overexpression [97], B-cell specific overexpression of BTK [98], or a deficiency in a range of signaling molecules, including the WASP, the SRC-family kinase member Lck/yes-related novel tyrosine kinase (LYN), or Fc-receptor γ2b [96,99]. On the other hand, spontaneous GCs can also originate from defects in T cells, such as the pathogenic accumulation of Tfh cells induced by IFNγ excess in the sanroque lupus model [100] or aberrant production of IL-17 [101]. Moreover, aberrant signaling in T cells has been implicated in autoimmunity, for example, a deficiency for the mTOR inhibitor Tsc1 [102] or a mutation in the phospholipase Cγ1 (PLCγ1)-binding site of linker for activation of T cells (LAT), downstream of the T cell receptor [103].

In spontaneous GCs, autoreactive B cells engage cognate T cell help and initiate loss of T cell tolerance via B-cell-intrinsic, MHC class-II-dependent antigen presentation and proinflammatory cytokine production. From analyses in WASP-deficient or BTK-overexpressing mice, a picture emerges in which B-cell-specific IL-6 production is critical to achieve cytokine and costimulatory signals that induce spontaneous GC formation [98,99,104,105]. The capacity of antigen-activated autoreactive B cells to engage cognate CD4^+^ T cells is facilitated by BCR, TLR, CD40, and IFNγ receptor (IFNγR) signals and involves various positive feedback loops. For example, BTK-overexpressing B cells promoted IFNγ production by T cells and showed high expression of IL-6 and surface CD86 expression, which was dependent on interactions with T cells [98,105]. Serum IFNγ levels are increased in SLE patients, already prior to clinical symptoms, coinciding with the appearance of autoantibodies [106]. A critical role for IFNγ is further supported by the finding that it can promote the development of antibody-secreting cells (ASC). Hereby, IFNγ synergizes with IL-2 and TLR7 ligands to induce epigenetic remodeling at the loci encoding interferon factor 4 (IRF4), B lymphocyte-induced maturation protein-1 (BLIMP1), and IL-21R [107]. B-cell-derived costimulatory signals were shown to be critical for complete Tfh cell differentiation, and even heterozygous deletion of CD80/CD86 was sufficient to prevent spontaneous autoimmune GC formation [108], further demonstrating the important role for the strength of B–T-cell interactions for proper regulation of the GC response. Whereas B-cell-intrinsic IFNγR, STAT1, and TLR7 signaling are essential for spontaneous GC formation in autoimmune B6.Sle1b mice, TLR9 has a negative regulatory function (see below) [109,110].

Interestingly, epigenetic modulation of GC B cells and ASCs can be influenced by metabolites derived from dietary fibers, such as the short-chain fatty acids butyrate and propionate. These metabolites were shown to decrease expression of AID and BLIMP1 in human and mouse B cells by upregulation of mRNAs [111] and to directly affect the epigenetic landscape at the *BTK* and *SYK* loci [112]. Although the role of the microbiome in pathogenic GC responses in autoimmune disease needs further investigation, these findings illustrate that environmental factors, such as the microbiota, may impact B cell activation.

### 5.4. Enhanced Plasmablast Differentiation

Not only GCs, but also extrafollicular responses are associated with SHM and CSR and can be involved in the formation of ASC-producing pathogenic autoantibodies. However, the contribution of each of these two pathways differs across patients and autoimmune disorders (reviewed by Malkiel et al. [113]). As described above, the expansion of autoreactive ASCs in SLE patients and mouse models does not appear to reflect defective antigen-specific tolerance but, rather, an overall plasma cell expansion. Because long-lived plasma cells do not respond to B cell depletion therapies, targeting these cells has been challenging. The high level of antibody production in plasma cells induces considerable endoplasmic reticulum (ER) stress. Consequently, plasma cells are very sensitive to proteasome inhibition, which leads to accumulation of misfolded proteins. Proteasome inhibition, which is widely used for the treatment of patients with plasma cell malignancies, was shown to reduce disease symptoms, plasma cell numbers, and autoantibody levels in various mouse models of SLE [114,115,116]. Favorable therapeutic effects of the proteasome inhibitor bortezomib were also observed in patients with severe/refractory SLE [117,118]. Enlargement of the ER is induced by the two key transcription regulators of plasma cell differentiation, BLIMP1 and X-box-binding protein 1 (XBP1), which also enhance mitochondrial mass and function, and thus promote oxidative metabolism [119]. Evidence was provided that mitochondrial dysfunction in B cells was associated with plasmablast differentiation and disease activity in SLE. In addition to ER stress, plasma cells also require a large amount of glucose, both as an energy source and for antibody glycosylation. Taken together, these findings imply that, next to the proteasome, XBP1, BLIMP1, and oxidative phosphorylation may also be potential therapeutic targets for autoimmune diseases.

### 5.5. Expansion of the Age-Associated B Cell Population

In many autoimmune diseases, including SLE, RA, SjS, and MS, an aberrant expansion of a specific B cell subset, commonly referred to as age-associated B cells (ABCs), has been described. These cells have a unique T-bet^+^CD11c^+^ phenotype in mice and humans and appear to be present in a preactivated state. ABCs efficiently produce proinflammatory cytokines, such as IFNγ and IL-6, have a high capacity to form ASCs, and develop rapidly into antigen-presenting cells. In SLE patients, these cells are major producers of autoantibodies and ABC accumulation correlates with disease activity [120]. ABCs can be detected both in peripheral blood and targeted organs. The unique expression profile of chemokine receptors, integrins, and myeloid markers enables ABCs to migrate to specific locations and to interact with cells in the micro-environment. The generation of ABCs is fueled by hyper-responsiveness to innate signals from endosomal TLR7 and TLR9, as well as adaptive signals, such as BCR engagement and T cell help via CD40/CD40L interaction, and IFNγ and IL-6 (Figure 1B, 6). The pathogenic characteristics of ABCs and their role in autoimmunity have recently been extensively reviewed [121,122].

## 6. B Cell Receptor Signaling in Autoimmunity

BCR signaling is directly linked to B cell survival, proliferation, differentiation, and effector functions. Dysregulation of BCR signaling as an important driver of autoimmunity is not only supported by genetic susceptibility associations with BCR signaling proteins and regulators, but also by efficacy of treatments that target signaling molecules in autoimmune animal models. An overview of the signaling pathways downstream of the BCR is shown in Figure 2.

### 6.1. Dual Role of LYN in BCR Signaling

The SRC family member LYN functions directly downstream of the BCR and can both promote and inhibit downstream signaling [123]. Phosphorylation of immunoreceptor tyrosine-based activation motifs (ITAMs) on the intracellular tails of CD79A/B (Igα/Igβ) initiates further downstream signaling through spleen tyrosine kinase (SYK), SLP65 (also known as the B cell linker protein BLNK), and Cbl-interacting protein of 85 kD (CIN85), which functions to oligomerize SLP65 and, thereby, poises cells for efficient initiation of downstream BCR signaling [124,125]. Phosphorylated SLP65 provides docking sites for BTK as well as PLCγ2, leading to Ca^2+^ mobilization and translocation of the nuclear factor κ-light-chain enhancer of activated B cells (NF-κB) and nuclear factor of activated T cells (NFAT) to the cell nucleus [126,127]. LYN and SYK also promote membrane recruitment and activation of phosphoinositide 3-kinase (PI3K), which results in activation of the protein kinase B (AKT) pathway, inducing B cell survival and proliferation [128,129].

Inhibition of BCR signaling is mediated by LYN through phosphorylation of inhibitory receptors, such as CD22, CD5, and FcγRIIB, which activate SRC-homology-region 2 (SH2)-domain-containing phosphatase (SHP-1) [130,131,132]. SHP-1 dephosphorylates LYN, SYK, and BTK, creating a negative feedback loop [133]. Furthermore, LYN promotes a direct feedback loop through activation of C-terminal SRC kinase (CSK), which inhibits activation of SRC family kinases [134]. FcγRIIB also inhibits PLCγ2, PI3K, BTK, and AKT signaling via SH2-domain-containing inositol polyphosphate 5-phosphatase 1 (SHIP-1) [135,136].

Due to redundancy within the SRC family, the initiation of downstream signaling is not dependent on LYN. In contrast, LYN does play an essential role in the negative regulation of BCR signaling [130,137,138]. Both complete and B-cell-specific LYN knock-out mice display a spontaneous SLE-like phenotype, featuring B and T cell activation, high serum ANA levels, and glomerulonephritis [139,140]. In addition to BCR signaling, LYN may also regulate signaling through TLRs, as deletion of MyD88 attenuates the autoimmune phenotype in *Lyn*^−/−^ mice, reducing type I interferon production and GC formation [141,142,143].

A critical role for LYN, SHIP, and CSK as negative regulators of BCR signaling in SLE would be supported by the finding of reduced expression or impaired activation of LYN and SHIP in B cells of SLE patients [144,145,146,147] and the identification of *CSK* as a genetic susceptibility locus [148].

### 6.2. Aberrant Levels and Activation of SYK, BTK, and PLCγ2 in Autoimmune Disease

In addition to LYN, altered levels or activation of other BCR signaling molecules have been found in various systemic autoimmune diseases. In SLE patients, a population of CD27^−^ B cells was identified that had increased expression of SYK protein and phosphorylation, both at baseline and upon BCR stimulation, and showed enhanced differentiation into IgG-producing cells [149]. In another study, enhanced SYK protein and phosphorylation were found in B cells from SLE patients, compared to healthy controls, and correlated with disease activity score [150]. Likewise, phosphorylated BTK (pBTK) and pPLCγ2 were increased in active SLE patients [150]. Moreover, in RA patients, the phosphorylation of SYK was enhanced in B cells, particularly in patients with high ACPA levels in serum, and could be reduced by targeting T cell costimulation with abatacept (a cytotoxic T-lymphocyte-associated protein 4 immunoglobulin fusion protein, CTLA4-Ig) [151]. Inhibition of SYK with fostamatinib in RA patients induced an improvement of symptoms compared to placebo [152,153], although these effects might, in part, arise from expression of SYK beyond B cells [154].

The role of BTK in the pathogenesis of autoimmunity has been studied extensively in animal models. *Btk* deficiency or therapeutic inhibition of Btk was protective in many rodent models of SLE and RA [155,156,157,158,159]. Conversely, increased expression of BTK specifically in B cells induced a spontaneous SLE/SjS-like phenotype in mice [98]. In human autoimmune disease, increased BTK protein levels and phosphorylation were found in B cells from ACPA^+^ RA, SjS, and glomerulonephritis with polyangiitis (GPA) patients with active disease [160,161]. Moreover, we found increased anti-Ig-induced phosphorylation of BTK and PLCγ2 in naïve B cells of patients with idiopathic pulmonary fibrosis, a chronic lung disease in which a pathogenic role is less evident [162]. Although autoimmunity may contribute to the disease phenotype, fibrosis is thought to be caused by an impaired healing response to recurrent micro-injuries. High BTK levels correlated with pathogenic T cell activation, and, similar to pSYK in RA, increased BTK protein levels were reduced upon abatacept treatment in SjS, suggesting that T cell interaction may be a driver of aberrant BCR signaling [160]. Inhibition of BTK by small-molecule inhibitors showed high efficacy in several preclinical autoimmune models and clinical efficacy was observed for fenebrutinib in RA and evobrutinib in MS [163,164]. Nevertheless, BTK inhibition also yielded diverging results in clinical trials [165]. We refer to Neys et al. [12] for a recent overview of various BTK inhibitors that are currently evaluated in clinical trials of various autoimmune diseases, including RA and SLE. Because BTK has kinase-independent functions [166,167], it might be more beneficial to target BTK protein levels. As BTK expression is known to be regulated by various micro-RNAs [168,169], it is attractive to target BTK through miRNA mimics [170].

Upon BCR stimulation, Ca^2+^ signaling in B cells is promoted through an interaction between B cell scaffold protein with ankyrin repeats (BANK1) and PLCγ2, which is enhanced by the SRC-family B lymphocyte kinase (BLK) [7,171]. Both BANK1 and BLK were identified as genetic risk loci for SLE and are functionally linked to type I interferon repression (Figure 2) [172]. Gain-of-function mutations in the *PLC*γ*2* gene in both human and mouse lead to a complex, severe immunodeficient and autoimmune phenotype [173,174,175]. In addition, pPLCγ2 levels are increased in B cells of active SLE and GPA patients [150,161]. However, because of the availability of a large range of well-tolerated BTK inhibitors with clinical efficacy in B cell malignancies, inhibition of BCR signaling as a therapeutic target for autoimmunity is mainly focused on targeting BTK [12], while PLCγ2 is currently not pursued.

### 6.3. PI3K–Akt–mTORC-Regulated Metabolism Is Essential for Normal B Cell Differentiation and Silencing of Autoreactive B Cells

Activation of PI3K induces the AKT–mTOR signaling pathway, which is negatively regulated by phosphatase and tensin homolog (PTEN), and controls cell survival and metabolism throughout B cell development and activation [176]. Balanced regulation of PI3K activity is critical, since patients with activated PI3Kδ syndrome (APDS) due to a PI3K gain-of-function mutation present with immunodeficiency and lymphoproliferation. Conversely, a subgroup of patients with common variable immunodeficiency (CVID) display disturbed BCR-activated PI3K signaling, particularly in ABCs [177].

Upon BCR-driven activation, naïve B cells switch from a metabolic dependency on fatty acids to glutamine-fueled mitochondrial respiration [178,179]. Additional switches occur during B cell differentiation, for example, as GCs progress. Hereby, the glycogen synthase kinase 3 (GSK3) acts as metabolic sensor that supports both the survival of naïve B cells and the generation and maintenance of GC B cells, which require high glycolytic activity [180]. Memory B cells are formed earlier in the GC response than long-lived plasma cells, and mammalian target of rapamycin complex 1 (mTORc1) expression and metabolism are lower in cells destined to become memory B cells, suggesting that temporal switches in the metabolic state may contribute to the differentiation fate [181,182].

Tightly regulated metabolism is crucial in the counterselection and silencing of autoreactive B cells throughout B cell development and activation. Metabolic reprograming through Glut-1 contributes to anergy of peripheral transitional B cells [178]. Anergic B cells remain metabolically quiescent upon stimulation, whereas chronically BAFF-stimulated B cells show rapid increased glycolysis, which is crucial for antibody production [178]. Increased BAFF levels, which are often present in autoimmune patients, may rescue autoreactive B cells from immune checkpoints and support the survival of anergic B cells [33,34,183]. Furthermore, it was shown in mice that GC B cells retain a hypoxic state that inhibits mTORc1 activity, promotes cell death, and limits proliferation and class switching to the proinflammatory IgG_2c_ isotype [184].

mTORc activity is increased in B cells from SLE patients and correlates with disease activity, plasmablast differentiation, and with B cell accumulation in salivary glands of SjS patients [185,186]. In SLE patients, mTOR-dependent autophagy is also increased in B cells, particularly in transitional and naïve B cells, and correlates with disease activity [187]. As described above, these B cell stages are subject to selection checkpoints, suggesting that increased autophagy may promote the escape of autoreactive B cells from central or peripheral tolerance [188]. Although, in autoimmune disorders, therapeutic approaches that alter B cell metabolism may be attractive, it will be challenging to develop cell-lineage-specific targeting strategies.

A critical role for the PI3K–AKT–mTORc would be supported by findings implicating micro-RNAs that regulate PTEN expression in autoimmune pathogenesis. PTEN expression is controlled by miR-148a, which is upregulated in SLE patients and lupus-prone mice and accelerates development of autoimmune disease in mouse models [189,190]. Although miR-148a regulates >100 genes, only a few target genes, including *PTEN*, drive its function in B cell tolerance [191]. In parallel, various micro-RNAs that limit PTEN expression were shown to control central B cell tolerance and to be dysregulated in B cells from patients with various autoimmune diseases [192,193,194,195,196,197]. Interestingly, antagonizing miR-7, which regulates PTEN expression, improved disease symptoms in MRL/*lpr* mice, signifying miR-7 antagonism as a potential treatment strategy in autoimmune disease [198]. Novel miRNA dysregulated in autoimmunity are continuously being discovered [199,200,201], many of which control key pathways in B cell activation, including CD40–CD40L interaction [202,203], TLR and type I interferon signaling [204], the GC response, and AID expression [205,206,207,208].

PTEN is also involved in the balance between IgM and IgD expression through upregulation of IgD [209]. It was recently shown that IgD levels on B cells determine the nature and duration of primary immune responses, with decreased levels of IgD leading to an accelerated but prolonged primary immune response and a delayed secondary response with lower levels of protective high-affinity IgM antibodies [210]. IgD has also been shown to attenuate anergy of transitional and mature self-reactive B cells [211]. Several lines of evidence point to a protective role of IgD expression in autoimmunity, including studies of IgD knockout in MRL/*lpr* mice, IgD transgenic mice, and treatment with activating anti-IgD antibodies in various autoimmune mouse models (reviewed by Nguyen et al. [212]).

## 7. Other Signaling Pathways Implicated in Autoimmunity

### 7.1. BAFF and APRIL as Drivers of B-Cell-Mediated Autoimmunity

BAFF signals through three different receptors, exerting differential effects during B cell differentiation: (i) naïve mature B cells require pro-survival signals through the BAFF receptor (BAFFR) [213,214,215]; (ii) negative regulation and class switch recombination are mediated through transmembrane activator and CAML interactor (TACI) [216,217,218,219]; and (iii) B cell differentiation and plasmablast or plasma cell survival are promoted through signals from a third receptor, B cell maturation antigen (BCMA) [220,221]. Many systemic autoimmune patients present with dysregulated BAFF levels in the circulation [222], and BAFF-overexpressing mice develop autoimmune pathology, resembling human SLE [34,97]. In SLE patients, soluble TACI and BCMA levels, but not BAFFR levels, are increased [223].

Binding of BAFF to BAFFR activates PI3K/AKT signaling in mature B cells, hereby regulating protein synthesis, metabolic fitness, and survival. The BAFFR activates the noncanonical NF-κB pathway but can also induce the canonical NF-κB signaling through crosstalk with the BCR, involving CD79A/B, SYK, and BTK (Figure 2) [224,225]. Interestingly, BAFF activates PI3K/AKT only in naive B cells [226]. BAFF-induced PI3K/AKT signaling requires direct interactions between BAFFR and BCR components CD79A/B and is enhanced by the AKT coactivator TCL1A. BCR expression levels are higher on the surface of naïve B cells than memory B cells, and IgM BCRs interact better with BAFFR than IgG or IgA, allowing stronger pro-survival responses from BAFF by naïve B cells. Furthermore, BCR signaling regulates BAFFR levels, and BAFF supports CD40 expression and T cell costimulation through BAFFR, suggesting the presence of a self-amplifying loop that supports the survival of self-reactive B cells in autoimmunity [227,228,229,230].

Signaling through TACI and BCMA is less well studied. TACI induces NF-kB, MAPK, and JNK activation through TRAF 2, 5, and 6, whereas BCMA activates NF-KB, AP-1, and NF-AT through TRAF 1, 2, and 3 [231,232]. TACI expression increases upon TLR9 stimulation [233], and, on marginal zone B cells, TACI interacts with TLR and mTOR signaling through binding of MyD88, together driving IgG class switching and antibody production [234]. TACI can be cleaved from B cells by ADAM10 and acts as a decoy receptor binding BAFF and APRIL, thereby blocking NF-kB activation and B cell survival [235]. Interestingly, increased soluble TACI levels in serum of SLE patients correlate with increased disease severity. Conversely, decreased expression of BCMA on B cells correlates with higher disease severity in SLE [236].

In addition to BAFF, APRIL also signals through TACI (with higher affinity) and BCMA (with lower affinity), thereby promoting IgA class switching and plasma cell survival, respectively [221,237]. APRIL levels in serum of SLE patients are increased [238] and associations with genetic polymorphisms in APRIL have been found [239,240]. Inhibition of BAFF or APRIL are being explored in autoimmune diseases (clinicaltrials.gov). Until now, only belimumab, which specifically targets BAFF, has been approved for treatment of SLE patients [36]. Although promising results were reported in SLE in early trials with atacicept, an IgG1 Fc–TACI fusion protein that binds BAFF and APRIL to inhibit TACI signaling, larger trials reported no clinical effect and increased risk of infection [241,242,243].

### 7.2. CD40–CD40L Costimulatory Signals and PTPN22 Downregulation in Autoimmunity

The interaction between CD40 and its ligand CD40L, which is highly expressed on activated T cells and Tfh cells, is critical for GC responses and for the formation of extrafollicular foci and antibody-secreting cells. Thus, it is evident that the CD40–CD40L axis is central to the pathogenesis of many autoimmune diseases, which is also supported by the identification of *CD40* as a susceptibility locus in SLE [244]. Blockade of CD40–CD40L interaction may, therefore, provide an opportunity for therapeutic application [245]. It has been reported that CD40 costimulation also downregulates the expression of the protein tyrosine phosphatase nonreceptor type 2 (PTPN2) and PTPN22 [246]. *PTPN22* is a major autoimmune risk locus: the R620W gain-of-function allele is found at high frequencies in patients with autoimmune disease, including T1D, RA, and SLE. Known functions of PTPN22 and their link to autoimmunity have recently been extensively reviewed [8,81,247]. It is established that PTPN22 is a negative regulator of SRC-family kinases and co-operates with CSK to inhibit BCR and TCR signaling (Figure 2), whereby the R620W variant interacts with CSK to a lesser extent. PTPN22 impacts BCR signaling in central and peripheral B cell tolerance, as well as activation of GC B cells and ABCs. However, the complete molecular mechanisms explaining the role of PTPN22 in autoimmunity remain unclear, particularly because it is not only expressed in lymphocytes, but in many other immune cells, such as macrophages, monocytes, and dendritic cells.

### 7.3. Inhibitory Co-Receptors of BCR Signaling Acting through SHP-1

As described above, upon BCR ligation, Lyn phosphorylates ITIM motifs on several inhibitory co-receptors that regulate BCR signaling through SHP-1. Depending on the ligand recognized by these inhibitory receptors and their unique expression profiles, they regulate activation of different B cell subsets with specific BCRs [248,249].

CD72 regulates BCR signaling upon binding of Sm/RNP small nuclear ribonucleoprotein particles and co-ligation with the BCR, thereby specifically regulating Sm/RNP-reactive B cells [250]. In addition, CD72 may also regulate TLR7-mediated activation upon Sm/RNP endocytosis, playing an important role in self-tolerance against nucleic-acid-containing antigens [250]. CD72-deficient mice develop a severe SLE-like autoimmune phenotype [251,252], and CD72 has been identified as an MRL gene involved in the autoimmune phenotype of MRL/*lpr* mice [252]. In SLE patients, CD72 expression levels on B cells are decreased [253] and, in children, this decrease is evident during disease flare but not in remission [254]. Furthermore, CD72 polymorphisms have been associated with SLE [255].

CD22 (or Siglec-2) is expressed exclusively on B cells and binds to α2,6-linked sialic acids, which are either present on the same cell (*cis*) or expressed by other cells (*trans*) [256,257,258,259]. Upon BCR activation, CD22 inhibits Ca^2+^ signaling in B-2 cells through activation of SHP-1 or GRB-2 (Figure 2) [256,260]. Inhibition of tonic BCR signaling by CD22 is restricted through interaction with the extracellular domain of CD45, which prevents CD22 function [261]. In contrast to the ligand-specific regulation of B cell activation by CD72, CD22-deficient mice show augmented regulation of Ca^2+^ signaling upon polyclonal BCR stimulation with anti-IgM [256,257,262]. However, depending on the genetic background strain, *Cd22*-deficient mice develop no or only a mild autoimmune phenotype [263]. Nevertheless, in SLE patients, *CD22* has been identified as a genetic susceptibility locus [264].

Another Siglec family member that may be involved in autoimmunity is Siglec-10 (Siglec-G in mice). In mice, Siglec-G regulates B-1 cells through SHP-1 [265,266]. This specificity for the B-1 subset may be due to recognition of α2,3-linked sialic acids in addition to α2,6-linked sialic acids [267]. In mice, Siglec-G has a protective role on an autoimmune background or collagen-induced arthritis [268,269]. In Guillain–Barré syndrome patients, polymorphisms in *SIGLEC10* have been identified that interfere with ganglioside recognition, which may hamper ganglioside self-tolerance in patients presenting with antiganglioside antibodies [270].

### 7.4. Inhibitory Co-Receptors of BCR Signaling Acting through SHIP-1

Another key phosphatase in the inhibition of BCR signaling is SHIP-1, which can be activated by several receptors. Peripheral tolerance of IgG BCRs is regulated by FcγRIIB, which crosslinks with the BCR upon binding of antigen-IgG immune complexes. This induces activation of LYN, which phosphorylates the ITIM on FcγRIIB, allowing subsequent recruitment of SHP-1 and SHIP-1 (Figure 2) [135,138,271,272]. Deficiency of FcγRIIB in mice leads to enhanced IgG humoral immunity and an SLE-like autoimmune phenotype [273,274,275]. In SLE patients, polymorphisms in the *FcγRIIB* gene have been associated with disease, and memory B cells fail to upregulate the expression of FcγRIIB [276,277,278]. This is more prevalent in African American patients, suggesting that dysregulation of FcγRIIB expression may, in part, explain the difference in ethnic susceptibility to SLE [278].

SHIP-1 can also be activated in an FcγRIIB-independent manner, which involves CD79A and LYN. The exact mechanism has not been fully elucidated, but SHIP-1 may directly interact with the ITAM on the intracellular tail of CD79A or interact with CD79A through adaptor proteins DOK3 and/or GRB-2 [279,280,281,282,283]. CD79A deficiency causes a developmental block at the immature B cell stage, although these cells do show enhanced signaling, suggesting a dual role for CD79A in BCR signaling [284,285]. In peripheral B cells, CD79A may play a role in the induction of B cell anergy through SHIP-1 activation, indicating a role in peripheral tolerance of self-reactive B cells [286,287,288].

### 7.5. A Pathogenic Crosstalk between B Cell Receptor and Toll-like Receptor Signaling in Autoimmune Disease

TLRs are expressed either on the cell surface or within endosomes and are crucial innate receptors recognizing pathogen-associated molecular patterns. TLRs can, however—in the context of autoimmunity—also be activated by endogenous ligands, such as self-nucleic acids. For example, due to impaired clearance of debris, such as necrotic cells and neutrophil extracellular traps (NETs), autoreactive B cells can be costimulated via TLRs. In this way, TLR activation is a potential pathogenic factor that can promote autoimmunity by stimulating antibody production, antigen presentation, and production of proinflammatory cytokines by autoreactive B cells. Mice lacking DNase1, the enzyme important for nucleic acid breakdown and important for NET clearance, develop an SLE-like phenotype [289]. Likewise, a decreased DNase1 activity has been described in SLE patients [290,291]. Reduced DNase1 activity can result in the accumulation of debris containing self-antigens, including histones and nucleic acids, to which autoantibodies are directed in systemic autoimmunity (Figure 3A).

An important role for TLR signaling in both initiation and progression of systemic autoimmune disease is supported by GWAS that identified risk genes involved in TLR signaling [292,293,294,295,296,297,298]. Pathogenic TLR stimulation can induce an autoimmune phenotype in various mouse models, including arthritis [299], experimental autoimmune encephalitis (EAE), a model for MS [300], and lupus [301]. Particularly TLR7, which recognizes single-stranded RNA in endosomes that is a typical feature of viral genomes, and TLR9, which recognizes unmethylated CpG sequences that are common in viral and bacterial DNA, are considered key players. A pathogenic role for TLR7 signaling has been shown in several mouse models [301,302,303,304,305,306,307] and, very recently, in SLE patients harboring *TLR7* gene mutations [308]. The TLR7^Y264H^ gain-of-function mutant was found to drive aberrant survival of BCR-activated B cells and accumulation of ABCs and GC B cells, resulting in a lupus-like phenotype, associated with aberrant survival of pathogenic autoreactive B cells in a GC-independent manner, suggesting an extrafollicular origin. This was in line with previous evidence that, in SLE, autoreactive B cells derive from extrafollicular responses through enhanced TLR7 responsiveness in combination with IL-21 and IFNγ and are poised to differentiate into ASCs [295,309]. The *TLR7* gene is located on the X chromosome and escapes X-chromosome inactivation [310]. As a result, TLR7 expression in pDCs, monocytes, and B cells from females is increased compared with men [311], which may, in part, explain the female bias in systemic autoimmune diseases.

In contrast, TLR8 and -9 signaling in B cells seem to function in a protective manner. Targeted deletion of TLR8 and/or TLR9 in several mouse models leads to a more severe autoimmune phenotype. Whereas this protective role for TLR9 was shown to be B-cell-intrinsic, there is only indirect evidence for TLR8 [307,312,313,314,315,316,317,318]. SLE patients display decreased TLR9 responsiveness, indicating an imbalance in TLR7 and 9 signaling in human systemic autoimmunity [319,320]. Both TLR7 and 9 compete for binding of Unc-93 homolog B1 (UNC93B1), a protein that regulates TLR trafficking from the ER to the endosomal compartment [321]. In addition, cessation of TLR7/9 signaling is mediated by the interaction between UNC93B1 and Syntenin-1. Mutations altering the function or interaction of these proteins can lead to systemic autoimmunity [322,323].

Engagement of TLR7 or TLR9 leads to receptor dimerization and subsequent recruitment of MyD88 to the intracellular Toll–interleukin receptor (TIR) domain (Figure 3B). This is followed by activation of IRAK4, IRAK1, and TNF-receptor-associated factor-6 (TRAF6) [324]. Further downstream, this leads to activation of TGFβ-activated kinase-1 (TAK1), the TAK1-binding proteins (TAB), and the p38/JNK/ERK and the NF-κB pathways. This enables translocation of transcription factors CREB, AP-1, IRF7, and NF-κB, stimulating survival and differentiation of B cells, as well as the production of proinflammatory cytokines, such as IL-6 and type I interferons (IFN-I). The interplay between BCR and TLR signaling, often in the context of autoimmunity, has been a topic of intense research (Figure 3B). First of all, BCR engagement enhances TLR expression [325,326,327]. Secondly, studies using transgenic mouse models show TLR4, 7, and 9 stimulation-induced B cell proliferation, survival, and cytokine production are significantly reduced or even absent when the BCR or SYK is lacking [328,329]. This TLR-mediated SYK activation was MyD88-independent and resulted in activation of the ERK and PI3K–AKT pathways. SYK was also shown to be indispensable for TLR9-induced B cell activation and differentiation in human B cells [246,330,331]. BTK interacts with the TIR domains of several TLRs [332] and with downstream signaling proteins, such as MyD88 adapter-like (MAL) [333]. The synergistic role of BTK in BCR and TLR9 signaling has been well described in murine and human B cells [334,335]. In addition to SYK and BTK, BANK1 enhances TLR signaling [336,337], whereas BCAP seems to modulate TLR signaling [338,339]. TAK1 was proven central to BCR–TLR synergy, as inhibition led to impaired B cell proliferation, differentiation, and cytokine production in response to combined BCR and TLR stimulation [340]. Dedicator of cytokinesis-8 (DOCK8) also links TLR stimulation to the BCR signaling cascade by inducing activation of SYK and STAT3 [341].

The type I IFN signature, a hallmark of several systemic autoimmune diseases, including SjS and SLE, involves a positive feedback loop including TLR signaling (Figure 3A). Plasmacytoid DCs (pDC) produce vast amounts of IFN-α in response to TLR stimulation, for example, following viral infection or in the excessive presence of apoptotic debris [342]. IFN-α stimulates TLR7 and MyD88 expression in B cells, leaving TLR9 expression unaltered [343]. Autoreactive B cells can take up self-antigens containing nucleic acids via endocytosis, where enhanced TLR7 expression can subsequently facilitate their pathogenic survival and differentiation. In turn, the production of autoantibodies is promoted. These can form immune complexes containing autoantibodies bound to self-antigens and nucleic acids, which can activate pDCs via FcγRIIA. Overall, this results in a vicious circle where increased TLR–BCR signaling leads to autoreactive B cell activation, which is thought to be important both during disease initiation and progression (Figure 3A,B).

Taken together, synergy of BCR and TLR7 stimulation promotes autoimmunity [344,345], whereas synergy with TLR9 stimulation enhances tolerance [76]. These findings stress the delicate balance in TLR signaling, indicate the importance of crosstalk with the BCR, and pave the way for potential therapeutic targets involved in both BCR and TLR signaling.

## 8. Concluding Remarks and Future Perspective

In this review, we focused on the key role of BCR signals in central and peripheral B cell tolerance checkpoints, as well as the interplay between the BCR and various other signaling pathways in antigen-activated B cells. It is clear that the full impact of aberrant signaling in the etiology of specific autoimmune diseases remains to be established. Research in this area is complicated by the fact that autoimmune disorders generally arise from additive effects of many common genetic risk variants and various environmental factors. Accordingly, the contribution of (i) an altered BCR repertoire, (ii) an increased sensitivity of naïve B cells for signals from the microenvironment, as well as inappropriate (iii) activation, selection, survival, or cytokine profile of B cells upon autoantigen encounter to disease pathology is different across diseases and individual patients. On the basis of the available knowledge on aberrant signaling pathways in autoimmunity, therapies have been developed, however, with variable efficacy.

Over the past few decades, mouse models for autoimmune diseases have provided a wealth of information and mechanistic insight into B cell signaling and have been of great value to unravel pathogenic pathways. This is particularly the case for SLE, as global defects in central or peripheral tolerance are often associated with ANA formation. This might be explained by the abundance of DNA and RNA molecules that are released upon cell death, which can activate B cells in a T-cell-independent manner. In contrast, it remains challenging to design models for tissue-specific autoimmunity, which are currently largely dependent on the exposure to specific protein autoantigens, such as collagen for RA and myelin oligodendrocyte glycoprotein in the experimental autoimmune encephalomyelitis model for MS. In this context, it is also of note that the nonobese diabetic (NOD) mouse, which is extensively studied as a model for T1D, also develops symptoms of SjS. As autoimmune diseases show multifactorial inheritance patterns, the generation of genetically engineered mouse models will benefit from gene editing tools that have the potential of simultaneous editing of multiple loci [346].

Although animal models will remain of great value, it is becoming increasingly clear that we are reaching limits as we gain more and more in-depth knowledge revealing critical differences in immune pathology between mouse and man [347]. Novel technology will be of great help to uncover factors that are critical for autoimmunity in humans in unprecedented detail. This is already clear from the impact of flow-cytometry-based techniques to study signal transduction pathways: phospho-flow cytometry is now used to measure and quantify phosphorylation of an ever-expanding list of critical B cell signaling proteins in conjunction with cell surface markers [348,349]. These methods allow a rapid and detailed analysis of small, distinct subpopulations of B cells at the single-cell level and provide a more quantitative read-out than classic Western blotting.

Another exciting development is that whole-genome sequencing of patients diagnosed with autoimmune disease has recently identified novel rare mutations. These mutations have provided evidence for a critical pathogenic role of various genes, including partial RAG deficiency [350] and gain-of-function mutations in the *IKFZ1* gene, encoding the Ikaros transcription factor [351] and the *TLR7* gene [308]. Single-cell technology will be instrumental to uncover drivers of interindividual variation in immune cells, which will help to interpret and prioritize risk variants identified by GWAS and to identify critical cell types in autoimmune diseases [352]. Epigenetic processes that determine the accessibility of genes and, thereby, their expression profile are more and more recognized as important factors. It is, therefore, encouraging that autoimmune risk variants, for example, for T1D, could be translated into mechanistic insights by the identification of (cell-specific) regulatory elements by single-cell epigenomics [353].

GWAS in autoimmune disease uncovered a number of critical susceptibility genes and loci, which have been consistently replicated or validated on the protein level in the past couple of years. However, the vast majority of risk-associated single-nucleotide polymorphisms (SNPs) identified in autoimmune disease are located in noncoding regions. Hereby, it often cannot be excluded that nearby SNPs, in high linkage disequilibrium with the identified SNPs, are in fact causal for the disease [354,355]. Risk-associated SNPs in noncoding regions are assumed to be located in regulatory elements, which might be quite distant from the genes they control. It is mostly elusive how SNPs affect gene expression, as they mostly act in a cell-type or activation-status-specific manner. Technology to study epigenetics, as well as various innovative computational tools that are now emerging will help to interpret and prioritize disease-associated SNPs [353,356,357].

At the same time, the obtained knowledge on altered epigenetic regulation in B cells or other cells of the immune system may open new avenues to predict disease outcome or design novel therapeutic strategies for autoimmune disease. Finally, given the promising results of BTK inhibition in RA and MS, it is expected that the field may also benefit from the ongoing discovery of a wide range of small-molecule inhibitors targeting critical signaling pathways in B cell malignancies.

## Figures and Tables

**Figure 1 cells-11-03391-f001:**
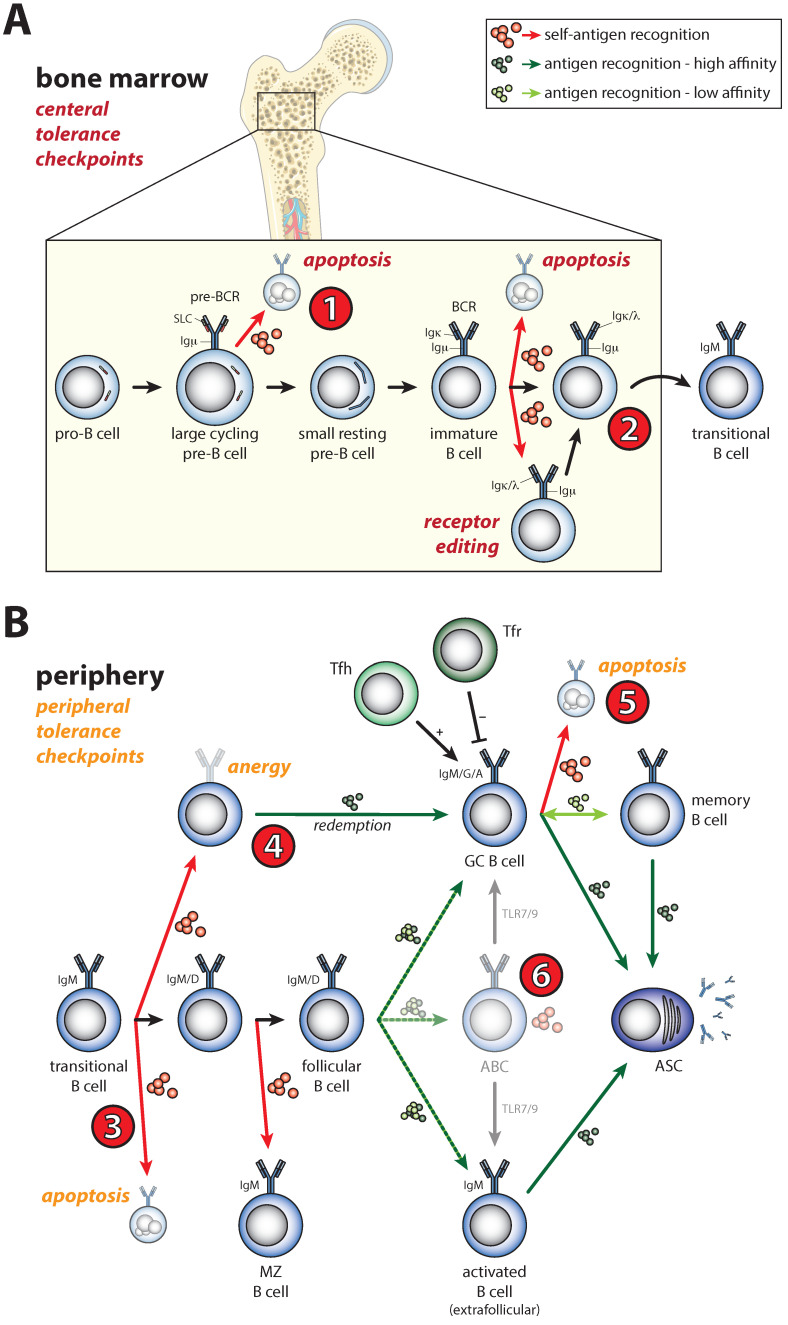
B cell selection is controlled by B cell receptor signals at several points during B cell development and activation. At several checkpoints during B cell development in the bone marrow (central tolerance) and B cell maturation and activation in the periphery (peripheral tolerance), autoreactive B cells can escape negative selection that is dependent on B cell receptor (BCR) signals. (**A**) (1) A functional pre-BCR will result in a positive selection and proliferation, whereas strong binding of self-antigens by the pre-BCR may induce apoptosis. At this stage, defective selection can result in the survival of B cells bearing a self-reactive BCR. (2) At the immature B cell stage, expression of a fully functional BCR, amongst other factors, results in the survival and positive selection of the B cell. However, expression of an autoreactive BCR should lead to either receptor editing or apoptosis. Defective selection at this stage can result in the escape of autoreactive B cells. (**B**) (3, 4) Transitional B cells emerging from the bone marrow are subject to the induction of apoptosis or anergy, when they recognize self-antigen. If defective, autoreactive B cells escape apoptosis or will not be constrained by anergy. (5) Germinal center B cells can undergo several rounds of selection and proliferation. During this process, somatic hypermutation generally increases affinity towards the antigen but can also generate B cells with self-reactive BCRs. Normally, such B cells undergo apoptosis, because essential survival signals, particularly those derived from activated T cells, are lacking. By contrast, in autoimmune disease, signals from the BCR may drive survival and differentiation of autoreactive B cells. (6) Various autoimmune diseases show an expansion of age-associated B cells that can be activated by TLR signaling and are prone to autoreactivity. See text for details. ABC: age-associated B cell; GC: germinal center; ASC: antibody secreting cell; MZ: marginal zone; Tfh: follicular T helper cell; Tfr: follicular regulatory T cell.

**Figure 2 cells-11-03391-f002:**
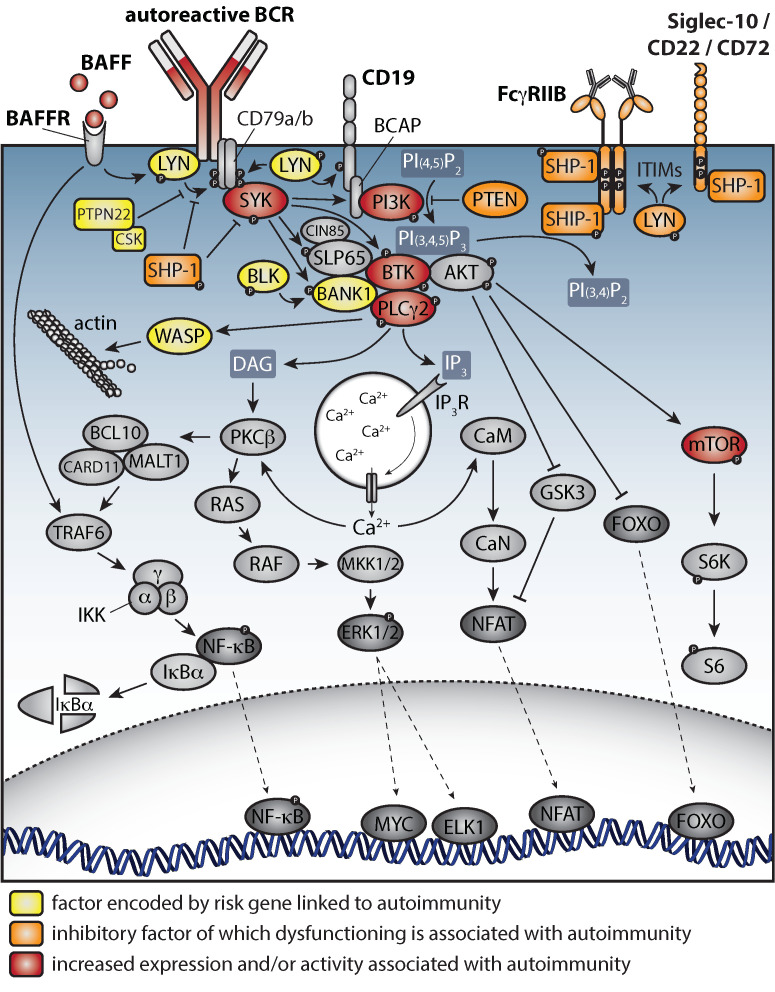
Signaling molecules downstream of the B cell receptor that are associated with autoimmunity. The signaling pathway downstream of the B cell receptor (BCR) contains signaling molecules that are partially shared with other pathways. In particular, the BAFF receptor provides essential survival signals and the CD19 co-receptor, which is intimately connected to the BCR. Inhibitory signals are provided by the kinase CSK and various phosphatases, including SHIP-1 and SHP-1, the activity of which is induced by Siglec-10, FcγRIIb, and CD22 surface receptors, as well as PTEN and PTPN22. Highlighted signaling molecules have been linked to an increased risk in the development of autoimmune disease (yellow), have crucial inhibitory function in BCR signaling by preventing autoimmunity (orange), or show increased expression and/or activity in B cells from autoimmune disease patients (red). Signaling molecule abbreviations are explained in the text.

**Figure 3 cells-11-03391-f003:**
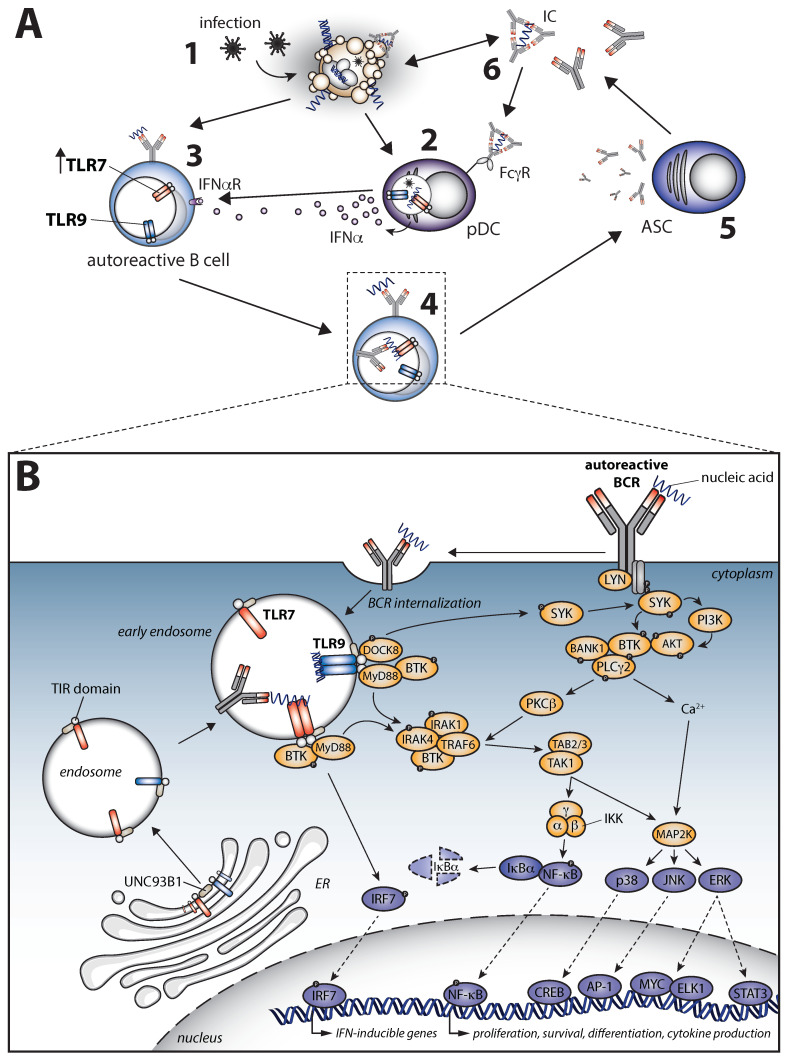
The vicious cycle of autoreactive B cell activation and the interplay between BCR and TLR signaling. (**A**) 1, A viral infection induces inflammation and cellular apoptosis and necrosis. This can lead to the accumulation of debris, containing autoantigens and nucleic acids. 2, Plasmacytoid dendritic cells (pDCs) are activated during viral infections via Toll-like receptors (TLR) and produce vast amounts of interferon-α (IFN-α). 3, IFN-α stimulates autoreactive B cells via the IRFα receptor, causing upregulation of TLR7 expression. Meanwhile, the B cell is activated via the B cell receptor (BCR) by self-antigen and internalizes the antigen. 4, The autophagosome, containing self-antigen bound to the BCR, and the endosome, containing TLRs, fuse. Self-antigens, containing TLR ligands, stimulate both BCR and TLR in a synergistic manner. 5, Autoreactive B cells proliferate and differentiate into antibody-secreting cells (ASC). These produce high numbers of autoreactive antibodies. 6, Autoreactive antibodies recognizing self-antigen form immune complexes (IC). In turn, these can promote inflammation at the site of deposition and stimulate pDCs via fragment crystallizable region γ receptor (FcγR) to increase IFN-α production, completing the cycle. (**B**) Synergy of BCR and TLR signaling pathways leading to the activation of autoreactive B cells. Signaling molecule abbreviations are explained in the text.

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
