# Peer review of "Aberrant B Cell Signaling in Autoimmune Diseases"

_cells, 2022, doi:10.3390/cells11213391_

Round 1

Reviewer 1 Report

In the presented review, authors summarize and discuss available experimental data on aberrant signal processing in B lymphocytes and its crucial contribution to the pathogenesis of autoimmune diseases. A special focus is given to signals emanating from the B cell antigen receptor and Toll-like receptors followed by a detailed meta-analysis of single molecules and pathways promoting autoimmunity. Overall, the presented article constitutes a comprehensive and in-depth review on the apparent and complex connections between B-lymphoid signaling elements and autoimmune processes. By directing their focus on individual signal transduction pathways, the authors provide valuable perspectives on the underlying causes and mechanistic participation of signal response networks to human diseases. The translational aspects of basic research are intriguing. Given its thorough and informative nature, this manuscript merits publication following minor modifications.

While the review fulfills its scope of collecting and presenting the current state of scientific knowledge, the manuscript would further benefit from discussing the experimental challenges and/or limitations (e.g. disease models, disease heterogeneity) of the past and present, and how those have been overcome and could further be approached in the future. In this context, it may also be informative to discuss why the majority of the reviewed data/conducted experiments stem from research on systemic lupus erythematosus and SLE-related mouse models and not to the same extent from other (B cell-mediated) autoimmune diseases. The detailed introduction of the biochemical signaling machinery in B cells is appreciated. Yet, the more recently discovered scaffolding element CIN85 should be included in the figure illuminating BCR signaling (Fig. 2).

Altogether, this excellently written manuscript will be very well received by the community.

Author Response

See submitted Word-file

Reviewer 2 Report

This excellent review by Corneth et al. covers extensively how aberrant signalling in B cells can lead and contribute to autoimmune diseases. The authors chose to focus on BCR and TLR signalling but also mentioned other immune receptors that may be involved on their own or in conjunction with the BCR/TLR pathways in these processes.

The review is very well written and beautifully illustrated. The references are accurate, up to date and relevant.

I only have 3 minor comments:

- Chapter “BAFF and APRIL in autoimmunity”: TACI and BCMA signalling are not described at all. Their differences to the BAFFR pathway could be at least cited particularly regarding their interaction with the BCR and TLR mediated axes. Moreover, although Belimumab is the only approved therapeutic targeting these axes in SLE, many others are under development and other approaches, particularly those based on Fc-recombinant proteins, could be cited.

-Chapter “Inhibitory co-receptors of BCR signaling acting through SHP-1”: Several papers have shown in a mouse model that Siglec-G was protective in SLE and could be cited (eg: Bökers et al JI 2014, Müller et al JI 2015).

-Chapter “Toll-like receptor signalling”: Although the papers are cited, there is no reference in the text to the potential protective role of Tlr8. This could be introduced to complete the part on Tlr9.

Author Response

See submitted Word file

Reviewer 3 Report

This review concerns an important topic and the authors have done a tremendous amount of work summarizing a large body of literature. The authors summarize literature pertaining to how B cell differentiation and signaling occurs and about how abnormal B cell signaling contributes to autoimmune disease development. Overall, this was quite densely written, and in it's current state, it read to me more like a book chapter than a review.

Strongly suggest reorganizing and focusing around the title and the abstract or reorganizing and making this a broader over view. As it stands, it is a massive review/book chapter. I congratulate the authors on their hard work summarizing a large body of literature, but to make an impact on the field, have some suggestions for improvement of the review as follows.

The abstract implies this review will be about TLR7 and TLR9 it perhaps over simplifies (and misleads) about TLR9 "enhancing tolerance" I can't tell if this relates to minor syntax issues or an oversimplification of the findings to date in the literature. In any case, the abstract does not reflect the broader review that follows in the text, starting with a very broad introduction about GWAS and then leading into figures that highlight normal B cell development signaling.

My specific comments for the authors are as follows

The title and abstract focus on signaling and endosomal TLRs does not reflect the body of the text in the manuscript. The review overall lacks "a voice."  The extensive review of B cell differentiation and normal B cell signaling (To include two figures) does little to synthesize the literature for the reader in the way the title and abstract suggest.

Figure 1.  The figure title suggests we will see how BCR signaling controls positive and negative selection. This could perhaps be made much clearer. As it stands the figure looks like every other figure about B cell differentiation and tolerance in the literature. While I appreciate that the authors put in self antigen and "antigen-recognition" symbols, these have little impact by eye. Perhaps revision of the figures to show how signaling influences differentiation of B cells toward an autoreactive phenotype. Certainly, the figure legend requires additional text and explanation of symbols etc.

Figure 2 - what is the main point of this figure as pertains to aberrant B cell signaling--this looks like normal BAFF and BCR and Signaling with many molecules depicted that are not described by the authors at pertinent to autoreactive B cells....I think the authors could better utilize this figure to highlight important facts from the text and then in the text, the reader could be "walked through" the figure. As written, this is a terribly difficult read and the figures do not enhance any major points the authors are trying to make. The figure legend does not reflect all that we see in the figure.

Along those same lines, the headers the authors have in italics might edited to better focus the review on aberrant B cell signaling.

Finally, if the authors are intending to address "various system and organ-specific autoimmune diseases" in this review, they might consider organizing the summary they provide around diseases.

Author Response

See submitted Word file

Round 2

Reviewer 3 Report

The authors have addressed my concerns